# *Alchemilla monticola* Opiz. Functional Traits Respond to Diverse Alpine Environmental Conditions in Karavanke, Slovenia

**DOI:** 10.3390/plants11192527

**Published:** 2022-09-27

**Authors:** Tadeja Trošt Sedej, Tajda Turk

**Affiliations:** Department of Biology, Biotechnical Faculty, University of Ljubljana, Večna pot 111, 1000 Ljubljana, Slovenia

**Keywords:** *Alchemilla monticola*, alpine plant, UV radiation, temperature, altitude, functional traits

## Abstract

Alpine plants are exposed to demanding environmental conditions, such as high ultraviolet (UV) and photosynthetic radiation, extreme temperatures, drought, and nutrient deficiencies. Alpine plants adapt and acclimate to harsh conditions, developing several strategies, including biochemical, physiological, and optical responses. However, alpine plants’ survival strategies are hardly researched due to time-consuming and complex experimental conditions, which are supported by scarce studies. Our study focused on the functional traits of the alpine plant *Alchemilla monticola* Opiz (hairy lady’s mantle) growing at two different altitudes (1500, 2000 m a.s.l.) and two different UV exposures per altitude. Near-ambient (UV) and reduced (UV-) UV radiations were provided by using two sorts of UV absorbing filters; temperatures were monitored hourly. The experimental plots were located at Tegoška Gora, Karavanke, Slovenia. Functional traits: physiological, biochemical, and optical characteristics were recorded three times during the growing season. *A. monticola* showed high maximum photochemical efficiency at both altitudes throughout the season, which confirms good adaptation and acclimatization of the plant. Furthermore, significantly higher maximum photochemical efficiency at the subalpine altitude coincided with significantly higher UV absorbing compounds (UV AC) contents at the subalpine compared to the montane altitude in August. *A. monticola* manifested high UV AC contents throughout the season, with significantly increased synthesis of UV AC contents in the subalpine conditions in August and September. The stomatal conductance rate increased with altitude and was correlated mostly to a lower temperature. *A. monticola* leaves did not transmit any UV spectrum, which corresponded to high total UV AC contents. The leaf transmittance of the photosynthetic spectrum increased at the subalpine altitude, while the transmittance of the green and yellow spectra increased under the reduced UV radiation in the autumn. *A. monticola*’s high photosynthetic spectrum transmittance at the subalpine altitude in the autumn might therefore be due to subalpine harsh environmental conditions, as well as plant ontogenetical phase.

## 1. Introduction

Alpine plants have developed several strategies to survive harsh alpine environmental conditions [1]. Alpine plants adapt and acclimate to high ultraviolet (UV) and photosynthetic radiation, large temperature flutter, drought, low air humidity, nutrient deficiencies, and a short growing season [2,3]. Alpine environmental conditions are stressful for plant life, where human activities also contribute to the stressful environment [3,4].

UV radiation plays a key role as an environmental modulator of plant metabolic regulation and growth [5]. UV-B (280–315 nm) has been found to alter leaf anatomy and morphology, resulting in smaller and thicker leaves, with thicker epidermis or cuticle [6,7]. Various plant species displayed increased palisade thickness when grown under increased UV-B [8,9], while leaf thickness increase in *Brassica carinata* and *Medicago sativa* has been attributed to an increased number of spongy parenchyma cells [10]. Studies reported increasing stomatal density [11] as well as increasing stomatal length [12] with increasing altitudinal gradient, which both positively correlated with increased transpiration rate.

Under high UV-B, plant growth and productivity can decrease [13,14,15,16], as well as the stomatal function [17]. Under ambient UV-B, photosynthesis and productivity might remain unchanged through effective photoprotection, which correlates with the biosynthesis of UV absorbing compounds (UV AC), a plant antioxidant protection system [18], and induction of photo repair [19].

UV-B effects on plants from high altitudes and low latitudes are usually indistinct, and they have a modulatory rather than a damaging effect [20]. Alpine plants have developed adaptation and acclimatization to high UV-B doses. Alpine plants might have co-tolerance to several stress factors; extreme temperatures, drought, and high photosynthetic radiation can also increase tolerance to UV-B. Such interactions can increase or decrease plant tolerance to the second stressor [21,22,23].

The present study investigated the functional traits’ responses of the alpine perennial plant *A. monticola* Opiz (hairy lady’s mantle) to changing UV radiation and temperature regimes at different altitudes in an alpine environment. The species is widespread from montane (800–1500 m a.s.l.) to subalpine (1500–2000 m a.s.l.) altitudinal zone, which indicates their successful adaptation and acclimation to the alpine environment. Research on *Alchemilla* encompasses phylogenetics [24] and medicinal plant studies [25,26] but not the ecology of the species, which is addressed in our study.

The alpine plant *A. monticola’s* functional traits were studied under different UV radiation doses (under near-ambient and reduced UV radiation) in its natural habitat at montane and subalpine altitude in the Julian Alps of Slovenia. Functional traits: physiological, biochemical, and optical characteristics were monitored three times during the growing season. We aimed to study the interactive effect of changing UV radiation and temperature regimes on the functional traits of *A. monticola* in its natural habitat.

## 2. Results

### 2.1. Maximum Photochemical Efficiency and Stomatal Conductance

*A. monticola* showed significantly higher maximum photochemical efficiency for the subalpine than the montane plants in August, with no changes in July and September. The stomatal conductance of *A. monticola* also significantly increased at the subalpine altitude in August, while it did not change according to the environmental conditions in July and September (Figure 1). 

### 2.2. Pigments Contents

*A. monticola* showed no significant changes in chlorophyll contents according to UV radiation and altitude throughout the season (Table 1). 

The UV-A AC and UV-B AC contents in leaves did not change according to UV radiation and altitude in July. However, significant increases in the UV-A AC and UV-B AC contents were measured for the subalpine altitude in August and September (Figure 2).

Temperature conditions as a proxy for altitudinal environmental factors significantly influenced the maximum photochemical efficiency of photosystem II, stomatal conductance rate, and UV AC in August, as well as UV AC in September. UV AC also responded to the interaction of temperature and UV radiation; meanwhile, chlorophylls did not respond significantly to any measured environmental conditions (Table 2).

RDA showed the influence of UV radiation and mean daily air temperature as a proxy for altitudinal environmental conditions on plant functional traits. The minimal and maximal temperatures explained less than 4% of the total variation of plant functional traits. In July, UV radiation and mean daily temperature accounted for only 2.2% of the total variation of plant responses, which can be explained as a consequence of a great difference in the ontogenetic development between montane and subalpine altitudes. In August, UV radiation and mean daily temperature accounted for 27.3% of the total variation of plant functional traits; in September, the two environmental parameters accounted for 11.3% of the total variation of plant functional traits (Figure 3).

### 2.3. Leaf Optical Properties

The leaf reflectance of *A. monticola* showed little changes according to UV radiation and altitude. The leaf transmittance of the total photosynthetic spectrum (except violet light) significantly increased under the reduced UV radiation (UV-) and at the subalpine altitude (Figure 4, Table 3).

## 3. Discussion

### 3.1. Plants Showed No Stress According to High Values of Maximum Photochemical Efficiency 

*A. monticola* showed no indications of stress according to the high values of its maximum photochemical efficiencies of PS II (Table 1). These showed an overall mean of 0.78 ± 0.05, which is characteristic of unstressed plants [27,28]. *A. monticola* exhibited significantly higher maximum photochemical efficiency under both UV radiation regimes at the subalpine altitude in August. Good constitutive and inducible defense strategy explains the effective protection of the photosystem, and consequently, the high maximal photochemical efficiency of *A. monticola* plants at the subalpine altitude.

The high maximum photochemical efficiency coincided with efficient constitutive and inducible defense compounds, accumulated as methanol-extractable UV AC, especially at the subalpine altitude (Figure 2).

Yamasaki et al. [29] showed that the major factor that contributes to thermal acclimation of photochemical efficiency is the plastic response of PS II electron transport to the environmental temperature. High UV and photosynthetic radiation can result in the decreased photochemical efficiency of PS II in weakly adapted and acclimated plants [30]. Plants have evolved several mechanisms to decrease the damaging effects of UV and photosynthetic radiation absorbed by the leaves’ pigments. Plant resistance to UV radiation depends on the formation of epidermal screening pigments and the repair processes [31].

The present results can be interpreted as being in good agreement with the research on higher plants [32], where UV screening was induced by low temperature even in the absence of UV-B radiation.

### 3.2. Stomatal Conductance Rate Responded to Temperature Conditions More Than to UV Radiation

*A. monticola* showed a significantly higher stomatal conductance rate at the subalpine than the montane altitude in August (Figure 1, Table 2). Studies have reported diverse effects of UV radiation on stomatal movements. UV-B can induce stomatal closure and thus reduce stomatal conductance [33], although it can induce stomatal opening [34,35,36]. Some studies exhibited increased stomatal conductance by the exclusion of solar UVB [37,38]. The research on trees exhibited that stomatal conductance increased with increased temperature [39].

Trošt Sedej et al. [40] reported that UV radiation increased stomatal conductance of *Saxifraga hostii* at high altitudes but not at low altitudes. *S. hostii* also showed trends for higher stomatal density and smaller stomatal length when exposed to the near-ambient UV, which corresponded to the higher stomatal conductance. Several studies have reported that stomatal conductance is related to total radiation, air and vapor pressure, and temperature [41]. Altitude, rather than UV radiation, affects the transpiration rates in rose cultivars. Plants grown at high altitudes had higher transpiration rates than plants grown at a lower altitude, regardless of the UV radiation [22]. 

The response of *A. monticola* affecting the stomatal conductance rate demonstrated that the temperature effect was more pronounced than the UV effect in high summer (Figure 3), when stomatal conductance increased with low temperature, but not in early summer and autumn, when the temperatures were lower, and leaf ontogeny was different. The complex *A. monticola* response exhibited the interactive effect of UV radiation and temperature on the stomatal conductance in diverse alpine environmental conditions.

### 3.3. Stable Chlorophyll Contents

*A. monticola* leaves manifested stable Chl a and Chl b contents under diverse UV radiation and temperature at the altitudinal gradient throughout the season (Table 1 and Table 2). Studies have shown that UV radiation might cause a decrease, an increase, or no change in chlorophyll content [42,43]. It has been established that UV-B radiation stimulates the biosynthesis of chlorophylls under a properly high intensity of photosynthetic and UV-A radiation [44]. UV-B also leads to free radical release, which results in chlorophyll degradation [45]. Variable photosynthetic pigment responses to UV-B are expressed between species or even cultivars [46]. Reduced UV-B radiation has an outcome in lowered chlorophyll contents in *Zea mays* and *Citrus aurantifolia* [47,48]. On the contrary, the chlorophyll contents increased in *Saxifraga hostii* under reduced UV-B radiation at montane and subalpine altitudes in September, which can indicate later senescence under reduced UV-B radiation [40]. High UV radiation can induce genes associated with senescence and consequently affect chlorophyll contents [49].

### 3.4. High UV Absorbing Compounds Contents throughout the Season

High methanol-extractable UV-A and UV-B AC contents were measured throughout the whole growing season in *A. monticola* (Figure 2). Plant UV protection is highly correlated to flavonoids, the most common UV AC, due to their UV absorbing and antioxidant properties [42]. Flavonoid formation is induced by UV radiation, as well as by environmental parameters, such as photosynthetic radiation, extreme temperature, water deficiency, and nutrient availability [43]. Researchers have demonstrated that plants from locations with naturally high UV radiation, at either high altitudes or low latitudes, are more tolerant to UV than plants from low UV radiation locations [50].

Some species modulate their UV screening properties within minutes to hours; these changes are driven by UV radiation [51]. In other alpine species, such as *H. nummularium* and *S. hostii*, the UV AC contents were mostly unaffected by the diverse environmental conditions, which indicated their high constitutive UV AC contents [40,52]. Constitutive levels of UV AC contents under increased UV radiation have been correlated with plant tolerance to UV radiation [53]. Constitutive and inducible UV AC contents reflect phenotypic plasticity in response to the different UV radiation. High inducible UV AC was found in low-altitude *Arabidopsis* ecotypes, while high constitutive UV AC was found in high-altitude ecotypes [54]. 

*A. monticola* responded to the subalpine environmental conditions with significantly increased synthesis of UV AC contents in August and September, which proved inducible UV AC contents. The stimulating environmental factor was mostly the low temperature at the subalpine altitude, as well as the synergistic effect of UV radiation and temperature (Table 2, Figure 3). The cross-response to UV radiation and temperature might play an important role.

### 3.5. The Leaves Did Not Transmit any UV Spectrum

*A. monticola* leaf reflectance expressed no significant differences due to the UV radiation and temperature, as a proxy for different altitudinal environment conditions, throughout the growth season (Figure 4, Table 3). A study of *S. hostii* confirmed that the leaf reflectance of the UV radiation and the photosynthetic spectrum were lowest for the near-ambient UV radiation at the subalpine altitude [40].

*A. monticola* leaves did not transmit any UV and violet spectrum (Figure 4, Table 3), which corresponded to its high constitutive and inducible UV AC contents (Figure 3). A similar lack of UV leaf transmittance was found in *Saxifraga hostii* [40]. Tropical alpine research indicated that high levels of UV AC can be common to native as well as to non-native plant species. The plasticity of the epidermal UV radiation transmittance is also common to some species. In the non-native *Verbascum thapsus*, leaf transmittance of UV-A was variable along an altitudinal gradient. In the native *Vaccinium reticulatum*, the leaf transmittance of UV-A was consistently low, unchanged with altitude [55]. In *Arnica montana,* the flavonoid content decreased with altitude; the triggering factor was temperature and not UV radiation [56].

*A. monticola* leaves transmitted more blue, green, yellow, and red spectra at the subalpine than at the montane altitude in September. Furthermore, the transmittance of the green and yellow spectra increased under the reduced UV radiation in September (Table 3). Leaf transmittance is correlated to the anatomical and biochemical features of the leaf [3] and is dependent on the plant life form [57]. The leaf optical properties are species specific and vary within a species due to its ontogeny [58]. *Saxifraga hostii* expressed earlier senescence with decreased photosynthetic pigments at alpine altitudes, which affected leaf transmittance in the photosynthetic spectrum [40]. Moreover, leaf optical properties vary according to environmental conditions, such as UV and photosynthetic radiation, temperature, and water availability [59]. *A. monticola’s* high photosynthetic spectrum transmittance at the subalpine altitude in the autumn might therefore be due to subalpine harsh environmental conditions, as well as plant ontogenetical phase.

## 4. Materials and Methods

### 4.1. Experimental Sites

The research site was located at Tegoška Gora in Karavanke, Slovenia. It consisted of montane and subalpine altitude plots (1500 m a.s.l.: 46°25′40.6″ N, 14°22′08.5″ E; 2000 m a.s.l.: 46°26′21.6″ N, 14°22′50.3″ E) (Figure 5). The research plots were on a southerly exposed slope above the timberline. The slope steepness was 80% to 100%. The vegetation was meadows on calcareous ground that lacked grazing and tourism. The mean annual precipitation of the site was 2500 mm to 2800 mm from the year 1991 to 2020 (Slovenian Environment Agency).

Two sorts of UV filters, Quinn XT UV (UV-), which absorbed UV-B and UV-A, and Quinn cast UVT (UV), which was transparent to UV-B and UV-A (Quinn-Plastics, Notts, UK), were placed above plants at the montane and subalpine plot, providing different UV exposures. Five UV and five UV- filters (20 × 20 cm) were installed 15 cm above a group of four to six plants at each altitude. The air and soil humidity near and under the filters was estimated as being comparable due to the small size of the filters and slope steepness. 

The share of UV-B radiation that reached the plants under the different UV filters was determined from UV-B radiation monitored over 5 days of a clear sky for all four different treatments. UV-B radiometer RM-22 was used (Opsytec Dr. Gröbel, Ettlingen, Germany). The mean daily UV-B doses (UV) and the reduced daily UV-B doses (UV-) were calculated over three months (July, August, September) and for the four treatments using a model [60] that included Caldwell’s generalized plant action spectra [61] for clear sky day measures. We assumed that the cloud cover and other meteorological phenomena at both altitudes were similar due to local proximity and uniform slope configuration (Table 4). 

Temperatures were monitored for the montane and the subalpine plot once per hour over three months (July, August, and September), using temperature data loggers HOBO TidbiTv2 (Onset Computer Corporation, Bourne, MA, USA) (Figure 6). The mean monthly differences between minimal (at 5 a.m.) and maximal (at 3 p.m.) temperatures at montane and subalpine plot are shown in Table 5. The data loggers were placed on the ground directly under the shade of plant leaves. 

### 4.2. Measurements

The alpine plant species *A. monticola* was analyzed three times during the growing season from July to September 2019. The UV filters were placed above the plants in mid-June, and the first sampling and measurements were taken 4 weeks after placing the UV filters, when the first leaves under different UV treatments developed. All measurements and analyses were performed on 10 randomly selected intact fully developed leaves from 10 plants per UV treatment at the montane and the subalpine altitude on clear sky days at around noon (photosynthetic photon flux density ≥1600 µmol m^−2^ s^−1^) on a second consecutive sunny day.

#### 4.2.1. Physiological Measurements

The maximum photochemical efficiency of photosystem II (*F_v_*/*F_m_*) at dark-adapted leaves was estimated with a fluorometer PAM-2100 (Walz, Effeltrich, Germany). The leaf stomatal conductance rate (g_s_) was monitored with a leaf porometer SC-1 (Decagon Devices, Pullman, WA, USA).

#### 4.2.2. Pigment Analyses

##### Photosynthetic Pigments 

The sampled leaves from 10 plants per UV treatment at each altitude were collected at noon, wrapped in moist paper, and kept in a refrigerated box. The fresh mass (FM) of the leaf tissue was extracted in 80% (*v*/*v*) acetone in buffered distilled water (pH 7.8). Absorbances were measured using a UV/VIS spectrophotometer Lambda 25 (Perkin Elmer, Akron, OH, USA). Chlorophyll a (Chl a) and chlorophyll b (Chl b) contents were calculated from the absorbances at 663.6 nm and 646.6 nm [62] and were expressed per leaf dry weight (mg g^−1^ DW).

##### Methanol-Extractable UV-B and UV-A Absorbing Compounds

The sampled leaves from 10 plants per UV treatment at each altitude were collected at noon, wrapped in moist paper, and kept in a refrigerated box. The FM of the leaf tissue was extracted in 5 mL of acidified MeOH (MeOH/H_2_O/HCl (37%), 79:20:1, *v*/*v*) [63]. Absorbances were measured over the spectral ranges for UV-B (280–315 nm) and UV-A (316–400 nm) using a UV/VIS spectrometer Lambda 25 (Perkin Elmer, Akron, OH, USA), were calculated per dry weight, and integrated to estimate the total content of methanol-extractable UV-B absorbing compounds (UV-B AC) and UV-A absorbing compounds (UV-A AC) (a.u. g^−1^).

#### 4.2.3. Leaf Optical Properties

The sampled leaves from 10 plants per UV treatment at each altitude were collected at noon, wrapped in moist paper, and kept in a refrigerated box. The leaf optical properties were determined in the laboratory on fresh material, using the Jaz Modular Optical Sensing Suite with a measurement sphere ISP-30-6-R applying UV/VIS/near-infrared light from a deuterium and halogen light source DH-2000 (Ocean Optics, Orlando, MA, USA). The leaf reflectance spectra were measured for the adaxial leaf surface. The spectrometer was calibrated to 100% reflectance using a white reference panel Spectralon (Labsphere, North Sutton, NH, USA) with >99% diffuse reflectance. The leaf transmittance spectra were measured for the abaxial leaf surface by illumination of the adaxial surface. The spectrometer was calibrated to 100% transmittance with a light beam that passed directly into the interior of the integrating sphere. The transmittance and reflectance measurements were processed by the SpectraSuite software (Ocean Optics, USA).

#### 4.2.4. Data Analyses

The plant responses and characteristic data were analyzed using the SPSS Statistics 22.0 software (IBM, Armonk, NY, USA). Statistical tests were performed on 10 samples. The normal distributions of the data were tested using Shapiro–Wilk tests. The homogeneity of variance was analyzed using Levene’s tests. One-way ANOVA with multiple comparison tests and Tukey post hoc tests were used to compare the differences between the UV treatments at the montane and subalpine altitudes. Two-way ANOVA was performed to investigate the effects of UV radiation and temperature, and their interaction, on the measured functional traits. Statistical relationships between the functional traits and environmental variables, UV radiation, and mean daily air temperature, were tested using a multivariate method for constrained ordination, redundancy analysis (RDA), performed by CANOCO 5.0. software (Microcomputer Power, Ithaca, NY, USA) [64].

## 5. Conclusions

*A. monticola* showed high maximum photochemical efficiency of PS II under the reduced and near-ambient UV radiation at the montane and subalpine altitudes throughout the season, which confirms good adaptation and acclimatization of the plant. Furthermore, significantly higher maximum photochemical efficiency at the subalpine altitude coincided with significantly higher methanol-extractable UV AC contents at the subalpine compared to the montane altitude in August. *A. monticola* manifested high methanol-extractable UV AC contents throughout the season, with significantly increased synthesis of UV AC contents at the subalpine conditions in August and September, which indicates additional inducible UV AC. High UV AC contents indicate phenotypic plasticity in response to variable UV radiation. The stomatal conductance rate increased with altitude and was correlated mostly to a lower temperature. The complex response of stomatal conductance exhibited the interactive effect of UV radiation and temperature in diverse alpine environmental conditions. *A. monticola* leaves did not transmit any UV spectrum, which corresponded to high methanol-extractable UV AC contents. The leaf transmittance of the photosynthetic spectrum increased at the subalpine altitude, while the transmittance of the green and yellow spectra increased under the reduced UV radiation in the autumn. *A. monticola’s* high photosynthetic spectrum transmittance at the subalpine altitude in the autumn might therefore be due to subalpine harsh environmental conditions, as well as plant ontogenetical phase.

## Figures and Tables

**Figure 1 plants-11-02527-f001:**
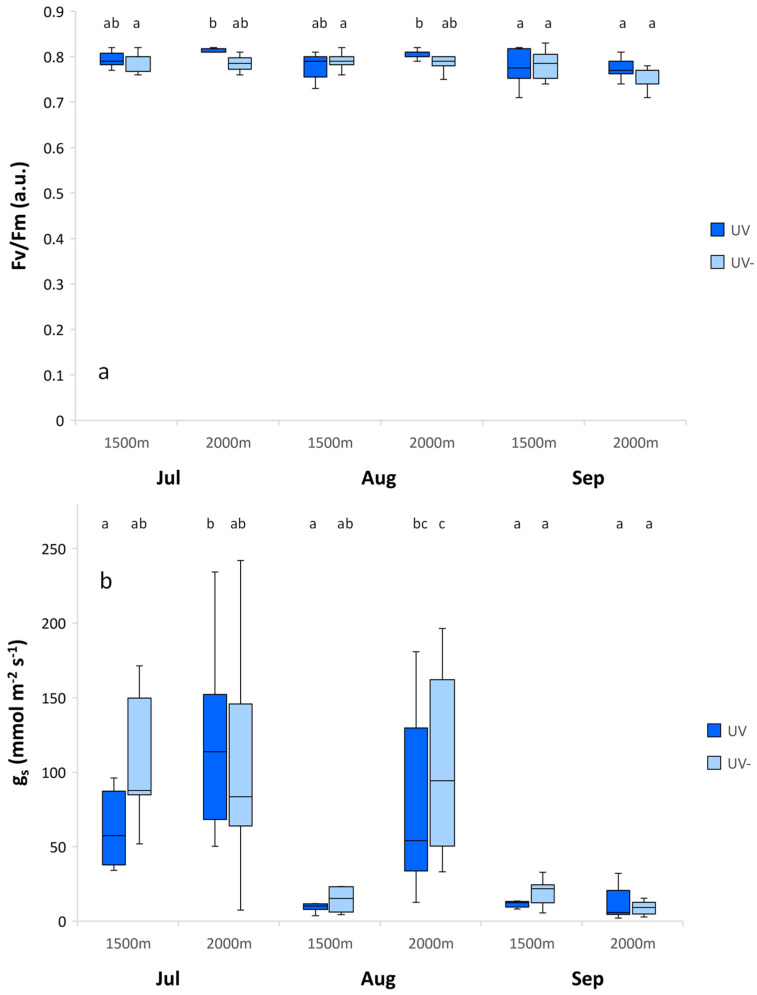
Maximum photochemical efficiency of photosystem II (*F*_v_/*F*_m_) (**a**) and stomatal conductance rate (g_s_) (**b**) for *Alchemilla monticola* according to month and altitude for near-ambient (UV) and reduced (UV-) UV radiation. Data are means ± standard error (*n* = 10 plants). Different letters indicate a significant difference between treatments in the same month (*p* ≤ 0.05; one-way ANOVA).

**Figure 2 plants-11-02527-f002:**
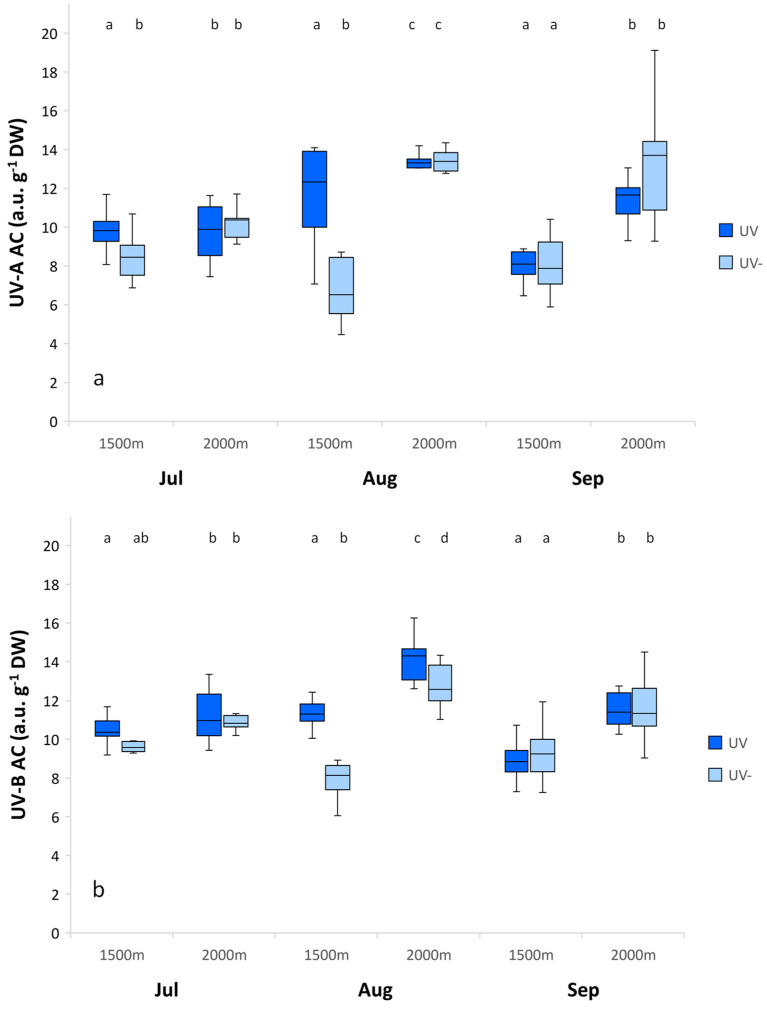
UV-A absorbing compounds (UV-A AC) (**a**) and UV-B absorbing compounds (UV-B AC) (**b**) contents of *Alchemilla monticola* according to month and altitude for near-ambient (UV) and reduced (UV-) UV radiation. (a.u.:arbitrary unit). Data are means ± standard error (*n* = 10 plants). Different letters indicate a significant difference between treatments in the same month (*p* ≤ 0.05; one-way ANOVA).

**Figure 3 plants-11-02527-f003:**
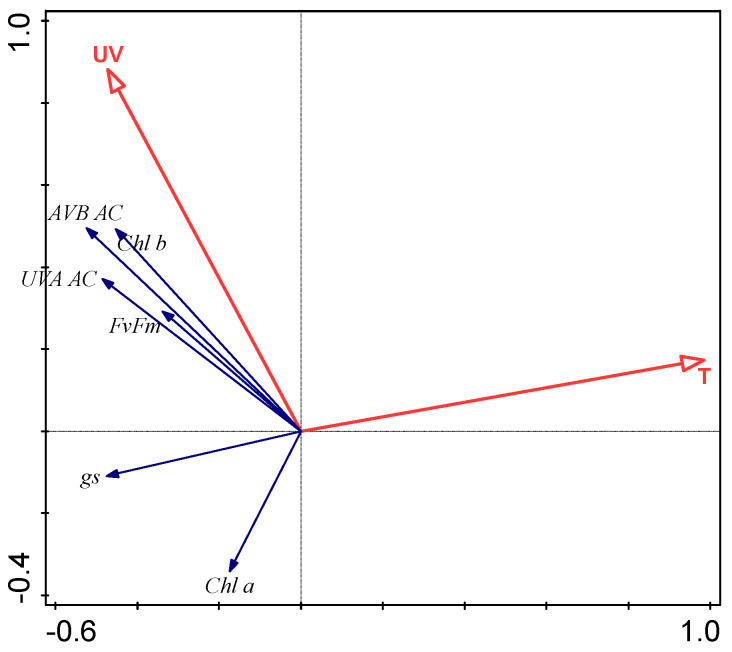
Redundancy analysis (RDA) biplot showing the strength of associations between temperature (T), UV radiation (UV), and plant functional traits: maximum photochemical efficiency of photosystem II (*F_v_*/*F_m_*), stomatal conductance rate (gs), chlorophyll a (Chl a), chlorophyll b (Chl b), UV-B absorbing compounds (UV-B AC), and UV-A absorbing compounds (UV-B AC) in August.

**Figure 4 plants-11-02527-f004:**
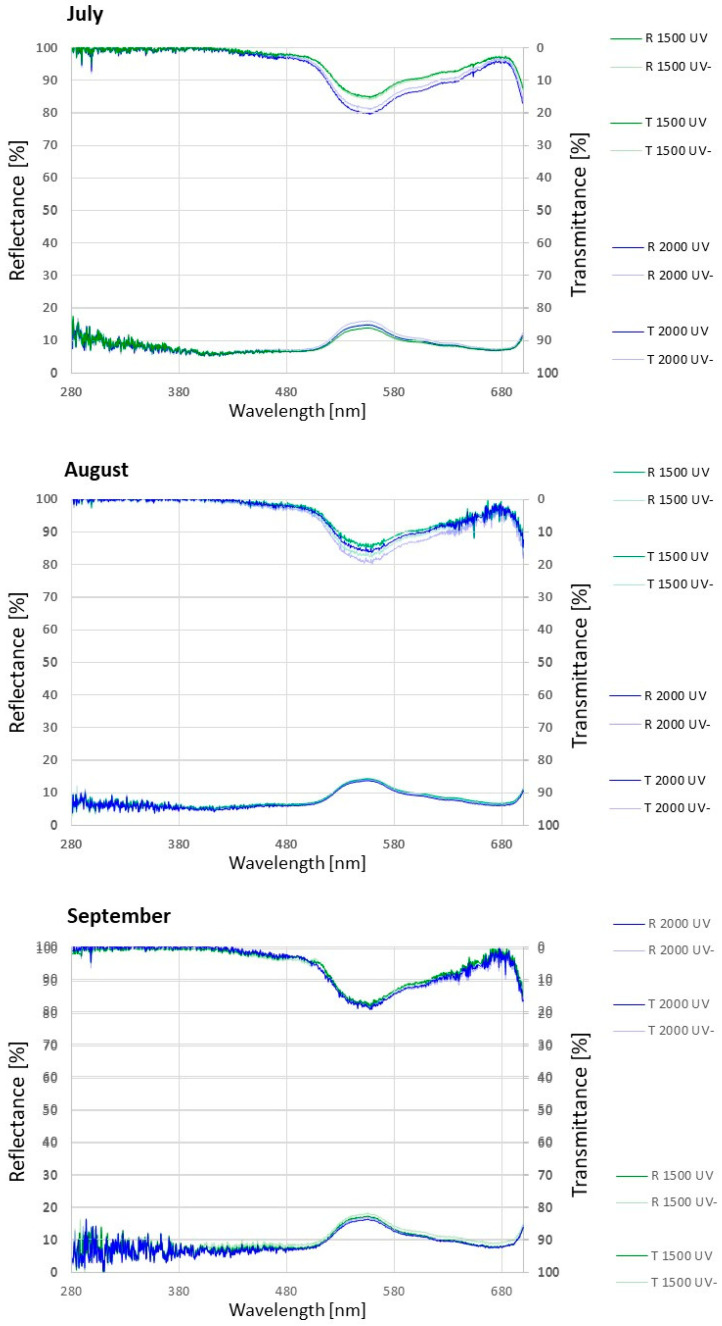
*Alchemilla monticola* leaf transmittance and reflectance in July, August, and September at montane (1500) and subalpine (2000) altitude under near-ambient (UV) and reduced (UV-) UV radiation. Data are means for every 5 nm interval (*n* = 10).

**Figure 5 plants-11-02527-f005:**
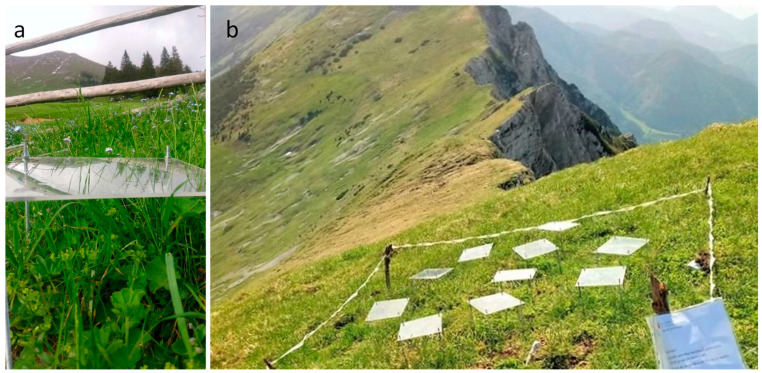
The two research plots of *Alchemilla monticola:* 1500 m a.s.l. (**a**) and 2000 m a.s.l. (**b**) at Tegoška Gora.

**Figure 6 plants-11-02527-f006:**
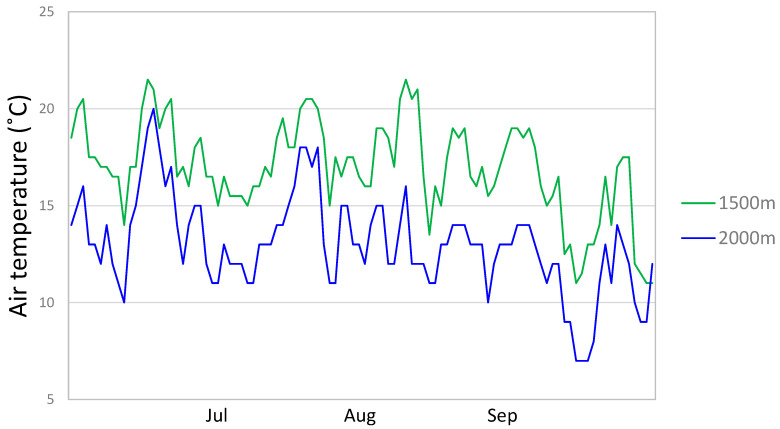
Mean daily air temperatures at 1500 m and 2000 m a.s.l. at Tegoška Gora, monitored once per hour in July, August, and September.

**Table 1 plants-11-02527-t001:** Chlorophyll a and Chlorophyll b contents of *Alchemilla monticola* according to month and altitude for near-ambient (UV) and reduced (UV-) UV radiation. Data are means ± standard error (*n* = 10 plants). Different letters indicate a significant difference between treatments in the same month (*p* ≤ 0.05; one-way ANOVA).

Measure	Month	1500 m		2000 m	
		UV	UV-	UV	UV-
Chlorophyll a (mg g ^−1^ DW)	July	7.55 ± 0.09 a	7.19 ± 0.21 a	6.46 ± 0.28 a	7.36 ± 0.35 a
Chlorophyll b (mg g ^−1^ DW)		4.40 ± 0.10 ab	3.79 ± 0.09 a	3.78 ± 0.11 a	5.33 ± 0.14 b
Chlorophyll a (mg g ^−1^ DW)	August	5.97 ± 0.11 a	5.43 ± 0.26 a	6.24 ± 0.16 a	6.15 ± 0.29 a
Chlorophyll b (mg g ^−1^ DW)		2.96 ± 0.18 ab	2.69 ± 0.11 a	2.84 ± 0.07 a	3.88 ± 0.09 a
Chlorophyll a (mg g ^−1^ DW)	September	7.42 ± 0.37 a	7.02 ± 0.25 a	7.24 ± 0.21 a	7.96 ± 0.36 a
Chlorophyll b (mg g ^−1^ DW)		6.52 ± 0.26 a	6.35 ± 0.25 a	6.46 ± 0.10 a	7.36 ± 0.23 a

**Table 2 plants-11-02527-t002:** Significance of response to environmental conditions: temperature (T), UV radiation (UV), and their interaction (T × UV) affecting maximum photochemical efficiency of photosystem II (*F*_v_/*F*_m_), stomatal conductance rate (g_s_), chlorophyll a, chlorophyll b, UV-A absorbing compounds (UV-A AC), and UV-B absorbing compounds (UV-B AC) of *Alchemilla monticola* in July, August, and September. Asterisks indicate a significant response to T, UV, and T × UV: (* *p* ≤ 0.05; ** *p* ≤ 0.01; *** *p* ≤ 0.001; ns *p* > 0.05; two-way ANOVA).

Measure				Two-Way ANOVA					
	July			August			September		
	T	UV	T × UV	T	UV	T × UV	T	UV	T × UV
*F*_v_/*F*_m_	ns	ns	ns	***	ns	ns	ns	ns	ns
g_s_	ns	ns	ns	***	ns	ns	ns	ns	ns
Chlorophyll a	ns	ns	ns	ns	ns	ns	ns	ns	ns
Chlorophyll b	ns	ns	ns	ns	ns	ns	ns	ns	ns
UV-A AC	ns	ns	ns	***	ns	ns	***	ns	*
UV-B AC	ns	ns	ns	***	ns	**	***	ns	*

**Table 3 plants-11-02527-t003:** Significance of response to environmental conditions: temperature (T), UV radiation (UV), and their interaction (T × UV) affecting leaf reflectance and transmittance of *Alchemilla monticola* over the light spectral ranges in September. Asterisks indicate a significant response to T, UV, and T × UV: (* *p* ≤ 0.05; *** *p* ≤ 0.001; ns *p* > 0.05; two-way ANOVA).

Spectral Range	T	UV	T × UV	T	UV	T × UV
	Reflectance			Transmittance		
UV-B	ns	*	ns	ns	ns	ns
UV-A	ns	ns	ns	ns	ns	ns
Violet	ns	ns	ns	ns	ns	ns
Blue	ns	ns	ns	*	*	ns
Green	ns	ns	ns	*	*	ns
Yellow	ns	ns	ns	*	*	ns
Red	ns	ns	ns	*	ns	ns
Near-infrared	ns	ns	ns	*	***	ns

**Table 4 plants-11-02527-t004:** Total and biologically active UV-B doses calculated for daily clear sky (UV) and reduced daily clear sky (UV-) UV-B according to month and altitude.

Filter	UV-B	UV-B Dose (kJ m^−2^ day^−1^)					
		July		August		September	
		1500 m	2000 m	1500 m	2000 m	1500 m	2000 m
UV	Total	55.91	57.58	47.75	49.27	34.52	35.73
	Biologically active	7.13	7.34	5.86	6.05	3.9	4.04
UV-	Total	16.77	17.28	14.32	14.78	10.36	10.38
	Biologically active	2.14	2.2	1.76	1.82	1.17	1.18

**Table 5 plants-11-02527-t005:** Mean monthly differences between minimal and maximal air temperatures at 1500 m and 2000 m a.s.l.

Δ T	July		August		September	
	1500 m	2000 m	1500 m	2000 m	1500 m	2000 m
(°C)	11.3	9.9	10.2	8.7	13.3	11.1

## Data Availability

The data presented in the study are available within the article.

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
