# Peer review of "Alchemilla monticola Opiz. Functional Traits Respond to Diverse Alpine Environmental Conditions in Karavanke, Slovenia"

_plants, 2022, doi:10.3390/plants11192527_

Round 1
Reviewer 1 Report
The manuscript deals with the effects of temperature, UV radiation under two altitudes of Alchemilla Monticola.
Authors have measured several biochemical, morphophysiological traits
The topic is intersting and meets te expectations of Plants.
Nevertheless, there are several concerns in this manuscript.
1- the experimental design is not presented
2-statistical analyses are limited. Indeed, multifactioral analyses (PCA for example) are lacking.
3- the most importnat concern is the approch developed in the manuscript. AUthors described results "simply" without any mechanistic approach which can relies physiological sequences (chronologically of functionally).
Author Response
Dear Reviewer.
I am grateful for the very constructive comments, which helped us to improve the manuscript. It brought new insight to the study.
Please, find the precise answers attached.
Best regards,
dr. Tadeja Trost Sedej

Reviewer 2 Report
The main idea of the study is interesting regarding the possible climate change implications in the behavior of alpine plants and it could fit perfectly with the topic of this special issue. In addition, field studies on alpine plants are not the most common studies found in literature due to the difficulties of carrying them out. The study showed that Alchemilla monticola alpine plants of Karavanke (Slovenia) are well-adapted to these harsh environmental conditions.
General concepts comments:
The methodological approach to studying the responses of Alchemilla monticola to UV radiation is not ideal from my point of view. Nowadays, the idea of UV radiation as stress in natural environments is changing to be considered more as a modulator since plants have adapted to environmental levels of this radiation. Therefore, the simple reduction using filters tends to have less impact on their behavior than, for example, a supplementation, which can help to analyze possible effects of a natural increase that could happen in the future.
In addition, some other weaknesses are listed below:
- Attending to UV methodological approach:
o As the author said, they used punctual measurements and extrapolated them to the whole study using a model, which does not take cloud cover into account. It could be a problem to compare two different sites, which can have differences, not only in general cloudiness but in fog or other meteorological phenomena such as the Foehn effect.
o In addition, due to the responsiveness of UVACs to different levels of UV, it would be necessary not only for the sampling day to be sunny but also for the previous days to be able to infer the influence of the modeled data. This information is missing from the manuscript.
- Regarding UVACs analysis:
o Not all UVACs are methanol-soluble, and their analysis must carry out a different extraction protocol. In addition, in some phylogenetic groups, these compounds, usually bounded at cell walls, have high importance in UV tolerance. It would be interesting to measure this other fraction in Alchemilla monticola to check its implication on UV tolerance.
o Although the content of global UVACs using spectrophotometry is a common technique to analyze the responses of plants to UV radiation, in some cases, a clear response in this global variable is not found but in the particular amounts of each compound. Although all of them are implied in the amount of UVACs, they could show a different antioxidant capacity to protect against oxidative stress. Analysis of individual compounds would highly improve the quality of the study. Measurements of the antioxidant capacity of the whole extract may also improve the manuscript.
- Other general concerns:
o The experimental design could be better. Natural conditions, especially temperature and radiation, are very labile variables, so the idea of inferring the data from a single day to a whole month is risky.
o References are most not very recent. If this is due to a lack of papers in montane and subalpine studies should be highlighted. In another case, more recent references should be used.
o The way to show the results is difficult to follow. The tables showed too many numbers that due to different samples and variables ordination makes it tough to read.
o The description of the results is very brief and omits part of the information reflected in the tables and figures.
Specific comments:
- Title: Consider changing the title. Although the authors mention just temperature as the factor that implies a response, other variables have to be considered even if the authors did not evaluate them. It is more appropriate the way they expressed in the last sentence of the abstract where they said 'alpine environmental conditions' or in the conclusions (line 317).
- Abstract: When the authors said “environmental condition” (line 8), “harsh condition” (line 10), and “environmental condition” (line 26); I think should be “conditions”.
- Abstract: line 26: remove “responses”.
- Introduction: line 31: Remove the capital letter from “Alpine”.
- Results: Paragraph 2.1: I missed some explanation for the difference in stomatal conductance rate (gs) found in 1500m + UV samples, especially in July. Why is it so low in comparison with the other treatments? and, Why did the authors omit an explanation?
- Table 1: I guess that (a.u) means arbitrary units but it must be explained in the legend.
- Results: Paragraph 2.2: Authors omitted the effect of the combination of Temperature and UV.
- Table 2: UVAC contents are expressed in a.u. g-1. It was fresh weight or dry weight? Specify, please.
- Table 2: In the contents of Chl B in 1500 m UV- September, the ± symbol is misplaced.
- Results: Paragraph 2.3: They did not describe the first two graphs of figure 1.
- Figure 1: I suggest using different color codes for transmittance and reflectance. It would make the figure easier to follow.
- Figure 1: Graph number 3. It seems in the axis there are two scales one above the other.
- Table 3 legend: It looks like there are two different font sizes in, the legend. Check please.
- Discussion: line 175. Change “deverse” by “diverse”.
- Discussion: line 201: This sentence has to be rewritten to be more clear.
- M&M: line 225: I guess montane should be written instead of mountain. In addition, add an “s” to plot.
- M&M: line 240, change the number 3 for “three”.
- M&M: section 4.2.2. Avoid text repetitions at the beginning of each subparagraph. (lines 262; 270; 278).
Author Response

(The authors gave the same response as above.)

Reviewer 3 Report
The study contributes to our knowledge about plant adaptation to harsh environmental conditions. However, before further consideration, several questions must be clarified.
First of all, it is not clear, which comparisons "a and b" in the Tables are reflecting, because even following the text descriptions doesn't the data on significance in Tables clearer.
Authors claimed that they showed that data were normally distributed before using ANOVA and SE. Could you please provide a supplementary table with a raw data? For me it seems unusual that ANOVA gives such a high significance for normally distributed data and so small overlapping SE.
Also some comments about Discussion. All UV AC part looks a bit simplified. What compounds are UV AC in the studied species? Do they only accumulate for the UV protections, or do they also have another biochemical functions, which might not be related to the UV levels?
I would like to point out that in the abstract and inroduction you mention morphological responses of studied plants, but nothing related to the morphology level is mentioned in the manuscript (unless you mean optic characteristics, which are not exactly morphological).
I also consider important to explain why no soil samples were taken at the experimental plots, ensuring the absense of other interfering factors but temperature, altitute, and UV.
Please find below other comments regarding this manuscript:
Sometimes you use "oxford comma" in your text, sometimes not - please, check this.
Lines 12-15 - better to divide this sentence as two.
Lines 18-19 - the sentence is not completely clear for me.
Line 61 - please change the reference to [ ] format.
Lines 60-61 - simple search in Google Academy shows almost 2000 results, including biochemical and morphological studies. This part must be improved.
Please add in the beginning of the Results section a brief description of the experimental scheme and specific parameters measured, along with mentioning of the methods (shorter version of the Section 4.1).
Figure 1 - What software was used to create the Figure?
Line 108 - please correct to "in September".
Line 109 - there is no "**" in the Table.
Lines 134-135 - please correct references.
Author Response

(The authors gave the same response as above.)

Reviewer 4 Report
In the manuscript entitled "Alchemilla monticola Opiz. functional traits respond to temperature conditions more than to UV radiation at two altitudes in the Slovenian Alps" the authors describe the effect of ultraviolet radiation on the physiological and biochemical processes of Alchemilla monticola. The topic of studying the effect of ultraviolet radiation on plants is very important and interesting, so this work has relevance. However, I cannot recommend this work for publication in the PLANTS journal.
There is a big question regarding the relevance and significance of this work, since there are no fundamentally new results. Also disappointing is the lack of modern molecular biological methods in the work, the use of which would help to reveal the mechanisms of adaptation of Alchemilla monticola to high mountain conditions.
I recommend using a shorter and more concise title of the work. For example: Influence of climatic conditions on growth and development Alchemilla monticola Opiz..
In the abstract, the authors claim to have studied the survival strategy of the alpine plant species Alchemilla monticola Opiz when exposed to UV radiation, which is generally not supported by the results. Before a complete study of the mechanism of adaptation, it is necessary for at least to study the work of a number of genes that are involved in this process.
I was unpleasantly surprised by the poor description of the results. The authors were able to describe all the results in 12 lines, this needs to be corrected.
Author Response

(The authors gave the same response as above.)

Round 2
Reviewer 1 Report
Thank you for considering positevily the recommendations done on the first version. The manuscript was greatly improved. It gained in clarty and scope.
Some typo errors reamin.
Author Response
Dear Reviewer.
We are grateful for constructive remarks on the present manuscript, which gained additional quality. The typo errors were corrected as well.
We hope that now you and other Reviewers find our manuscript of suitable quality and reader interest for publication in Plants.
Best wishes,
Dr Tadeja Trost Sedej

Reviewer 2 Report
Thanks for your editing work.
The main concerns were assesed and the changes in the text and figures have improved the paper. Data are much clear now in the figures.
However, I still think that the methodological approach should be improved for next experiments, including more precise measurements and further research about indicidual UV absorbing compounds.
Have a good day and good luck with your future projects.
Author Response
Dear Reviewer.
We are grateful for constructive remarks on the present manuscript, which gained additional quality. Definitely, we will improve the methodological approach to UV-absorbing compounds in our further studies.
We hope that now you find our manuscript of suitable quality and reader interest for publication in Plants.
Best wishes to you in every aspect of your life path.
Yours sincerely,
Dr Tadeja Trost Sedej

Reviewer 3 Report
As I see, the authors improved the manuscript and took into account all the points raised. They also provided a raw data, which is a a good step towards open science.
Author Response
Dear Reviewer.
We are grateful for constructive remarks on the present manuscript, which gained additional quality.
We hope that now you find our manuscript of suitable quality and reader interest for publication in Plants.
Best wishes,
Dr Tadeja Trost Sedej

Reviewer 4 Report
Thanks to the authors for the work done and the response to the comments. After reviewing this version of the work and the response of the authors, I have only a small number of comments and questions. In this version, the authors made specific emphasis on the significance of this work, and also presented the results in a more presentable way. On the whole, I can recommend this manuscript for publication after minor corrections.
Fig 1 and 2. Label the drawings as A and B, and also bring the captions to the drawings in order.
Fig 2. Are the UV-B Aug 2000m values valid?
Author Response

(The authors gave the same response as above.)
